# Effect of Purslane (*Portulaca oleracea* L.) on Intestinal Morphology, Digestion Activity and Microbiome of Chinese Pond Turtle (*Mauremys reevesii*) during *Aeromonas hydrophila* Infection

**DOI:** 10.3390/ijms241210260

**Published:** 2023-06-17

**Authors:** Shiyong Yang, Langkun Feng, Jiajin Zhang, Chaozhan Yan, Chaoyang Zhang, Yanbo Huang, Minghao Li, Wei Luo, Xiaoli Huang, Jiayun Wu, Xiaogang Du, Yunkun Li

**Affiliations:** 1Department of Aquaculture, College of Animal Science & Technology, Sichuan Agricultural University, Chengdu 611130, China; yangshiyong@sicau.edu.cn (S.Y.); f975970587@163.com (L.F.); 18279284293@163.com (J.Z.); yan18282257303@126.com (C.Y.); zcy27219542@163.com (C.Z.); 13228149625@163.com (Y.H.); liminghao1013@126.com (M.L.); luowei@sicau.edu.cn (W.L.); hxlscau@126.com (X.H.); 2Department of Engineering and Applied Biology, College of Life Science, Sichuan Agricultural University, Ya’an 625014, China; abc2004wjy@163.com (J.W.); duxiaogang@sicau.edu.cn (X.D.)

**Keywords:** *A. hydrophila*, purslane, Chinese pond turtle, intestine, microbiome

## Abstract

Large-scale mortality due to *Aeromonas hydrophila* (*A. hydrophila*) infection has considerably decreased the yield of the Chinese pond turtle (*Mauremys reevesii*). Purslane is a naturally active substance with a wide range of pharmacological functions, but its antibacterial effect on Chinese pond turtles infected by *A. hydrophila* infection is still unknown. In this study, we investigated the effect of purslane on intestinal morphology, digestion activity, and microbiome of Chinese pond turtles during *A. hydrophila* infection. The results showed that purslane promoted epidermal neogenesis of the limbs and increased the survival and feeding rates of Chinese pond turtles during *A. hydrophila* infection. Histopathological observation and enzyme activity assay indicated that purslane improved the intestinal morphology and digestive enzyme (α-amylase, lipase and pepsin) activities of Chinese pond turtle during *A. hydrophila* infection. Microbiome analysis revealed that purslane increased the diversity of intestinal microbiota with a significant decrease in the proportion of potentially pathogenic bacteria (such as *Citrobacter freundii*, *Eimeria praecox*, and *Salmonella enterica*) and an increase in the abundance of probiotics (such as uncultured *Lactobacillus*). In conclusion, our study uncovers that purslane improves intestinal health to protect Chinese pond turtles against *A. hydrophila* infection.

## 1. Introduction

Chinese pond turtle (*Mauremys reevesii*) is a widespread species that mainly inhabits lakes, rivers and ponds in East Asia [1]. It has now become one of the most widely cultured turtle species in China because of its high commercial value [2]. However, with the popularity of turtle breeding and the increase in breeding density, various bacteria and viruses have come along and caused serious hindrances to the development of the turtle breeding industry [3]. *Aeromonas hydrophila* (*A*. *hydrophila*) is a rod-shaped Gram-negative bacterium and can infect humans, livestock and aquatic animals [4,5]. In turtle breeding, *A. hydrophila* infection causes more than 15 conditions, such as red neck disease, septicemia, furunculosis, etc., which accounts for about 60% of the total disease cases in turtles [3]. However, little is known about the effects of infection with *A. hydrophila* on the intestinal tract of Chinese pond turtles.

The intestine, a very important digestive organ, has also evolved into an ecosystem in which a large number of microbes live [6,7]. Unlike the outer environment, the intestine provides a stable living place for microbes [8]. In return, the microbiota supplies the intestine with various metabolites to maintain its metabolic balance and confer resistance to infection [9]. Therefore, the intestine and its microbiota are commonly considered as a whole and any factors that affect the microbiota of the intestine can cause physiological dysfunction in the body [10]. Researchers found that *A. hydrophila* infection could alter the composition of intestinal microbiota in aquatic animals, such as in Chinese sea bass [11], grass carp [12] and blunt snout bream [13], resulting in disordering their intestinal function. To protect intestinal health, antibiotics are widely used to prevent pathogenic infections in aquaculture. However, overuse of antibiotics not only affects environmental safety, but also increases various risks to human health [14]. In addition, a study on zebrafish found that antibiotics in the environment increase the abundance of harmful bacteria in their gut [15]. Consequently, there is an urgent necessity to determine environment-friendly additives to replace antibiotics, in this case, plant extracts have been extensively studied and are considered to be the most promising alternative nowadays [16].

Purslane (*Portulaca oleracea* L.) is known as one of the most commonly used medicinal plants by the World Health Organization and has been described as the term “Global Panacea” [17,18]. Purslane simultaneously possesses multiple bioactive functions, such as antibacterial, anti-inflammatory, antioxidant, analgesic and wound-healing properties, showing a wide range of potential for the treatment of diseases [19,20]. In aged rats [21], purslane polysaccharides significantly promote the growth of probiotics and inhibited the reproduction of pathogenic bacteria in the intestinal tract. Studies in ceca of broilers showed that purslane significantly reduced the abundance of *E. coli* and significantly elevated the levels of *Lactobacillus* and *Bifidobacterium* [22]. Zhang et al. [23] found that purslane could ameliorate intestinal inflammation in mice by regulating endoplasmic reticulum stress and autophagy. These studies indicate that purslane improves the health of the intestine and its microbiota. However, the protective function of purslane on the intestine of Chinese pond turtles infected with *A. hydrophila* is unknown, therefore, this experiment aims to explore the effect of purslane on the intestinal structure, digestibility and microbes of diseased Chinese pond turtles, and provide a reference for the application of purslane in the treatment of bacterial infectious diseases in aquacultural animals.

## 2. Results

### 2.1. Purslane Mitigates the Symptoms and Increases Survival and Feeding Rates of Infected Turtles

To explore the protective function of purslane on turtles during *A. hydrophila* infection, the observable symptoms on the body surface were recorded, and survival and feeding rates were analyzed. The results showed that the body surface was intact and no visible damage was observed in the control group (Figure 1A), while the challenge with *A. hydrophila* caused epidermal ulceration of the limbs and exposure of the muscle layer of the turtle (Figure 1B). Compared to the infection group, the treatment by purslane showed neonatal black epidermis on the ulcers of the extremities (Figure 1C). The survival rate analysis showed that the turtles started to die from day 8 and died entirely on day 11 in the infection group. In the purslane group, the turtles also appeared dead on day 8, but most of them were still alive on day 11, and the cumulative survival rate was 35% (Figure 1D). The feeding rates of groups were presented by a heat map, which displayed a quite low feeding rate in the infection group (Figure 1E). By contrast, the feeding rate was obviously higher in the purslane group than in the infection group.

### 2.2. Purslane Improves the Intestinal Morphology and Digestive Enzyme Activities of Infected Turtles

Hematoxylin and eosin (H&E) stain of the intestinal tract was performed to further evaluate the protective effect of purslane on the turtles during *A. hydrophila* infection. The results showed that turtles had neatly arrayed intestines villi in the control group with highly intact mucosae and normal morphological structure (Figure 2A,D). However, *A. hydrophila* infection caused shedding and fracture of the intestinal villi, necrosis of intestinal epithelial cells, the blurred outline of villus, and more inflammatory cells infiltrated in the intestinal lumen (Figure 2B,E). After the treatment with purslane, the villus morphology was repaired, the number of shed mucosae in the intestinal lumen was decreased, and the number of neonatal epithelial cells in the mucosa was increased (Figure 2C,F). Statistics on villus height and crypt depth showed no significant difference among groups (Figure 2G,H). Intestine index analysis displayed significantly high pathological damage in the infection group (*p* < 0.05), while purslane significantly lowered intestinal damage (*p* < 0.05) (Figure 2I).

Here, we also assayed the activity of three digestive enzymes, containing α-amylase, lipase and pepsin. The results showed that their activities were obviously decreased after the infection but markedly rescued by purslane (*p* < 0.05) (Figure 2J–L).

### 2.3. Global Analysis of OTUs

The 12 samples from three groups underwent microbiome analysis. A total of 651,582 valid sequences with an average length of 414 bp were obtained (Table 1). OTU levels among different groups were analyzed to compare the species’ distinctiveness (Figure 3). The results showed that the community composition among the three groups presented distinct diversity. A total of 1354 OTUs were identified from all the samples (similarity: <97%), but only 166 OTUs were shared among the three groups (Figure 3A). Ternary phase diagram analysis indicated that these OTUs of the three groups belonged mainly to the phylum Bacteroidota, Proteobacteria, Firmicutes, Cyanobacteria and Desulfobacterota (Figure 3B). The phylum Bacteroidota and the phylum Proteobacteria were found primarily in the *A. hydrophila* infection group, while the phylum Firmicutes was the microbiota with higher abundance in the purslanes group (Figure 3B). The raw 16S rRNA sequencing data have been submitted to the Sequence Read Archive (SRA) database of the National Center for Biotechnology Information (NCBI), with accession number PRJNA957584.

### 2.4. Purslane Alters the Diversity and Richness of the Intestinal Community of Infected Turtles

The diversity and depth of microbial community structure were assessed to analyze the effect of purslane on the intestine of Chinese pond turtles infected with *A. hydrophila*. Rank-abundance and dilution curves that illustrate the difference in community abundance and evenness among the samples, respectively, did not show significant differences (Figure 4A,B). The Shannon’s index and Simpson’s index were used to compare community diversity among samples, which showed that the diversity of the purslane samples > control > infection samples (Figure 4C,D). Ace’s index and Chao’s index were used to evaluate community richness, and the results showed that purslane significantly enhanced the number of community species (Figure 4E,F). 

Beta diversity analysis showed that the richness and evenness of the microbial communities in purslane and control samples were higher than in infection samples (Figure 5). The microbial communities were dominated by Clostridiaceae, Lachnospiraceae and Peptostreptococcaceae in the purslane samples, but in the control group, Lactobacillaceae and Weeksellaceae were the main microbiota (Figure 5). In contrast, the dominant microbiotas of the infection group were Enterobacteriaceae and Erysipelatoclostridiaceae (Figure 5). Principal component analysis (PCA) displayed a marked difference in the microbial community structure of different groups, but high repeatability within each group (Appendix A).

### 2.5. Effect of Purslane on the Abundance of Intestinal Microbiota of Infected Turtles

In the sunburst plots, Firmicutes, Proteobacteria, Bacteroidota, Cyanobacteria and Desulfobacterota were the top 5 most microbes in the three groups (Figure 6A). The most microbes in the control, infection, and purslane groups were Firmicutes (48.5%), Proteobacteria (42.1%), and Firmicutes (80.2%), respectively. The pie charts and bar diagrams were used to further show the abundance of each group of microbiotas, respectively (Figure 6A,B). At the genus level, the most microbes in the control, infection, and purslane groups were *Lactobacillus* (21.19%), *Citrobacter* (24.52%), and *Clostridium* (36.42%), respectively. At the species level, the most microbes in the control, infection, and purslane groups were uncultured *Lactobacillus* (20.57%), *Citrobacter freundii* (24.52%), and uncultured *Clostridium* (15.64%), respectively.

### 2.6. Purslane Reduces the Proportion of Potential Pathogens and Raises the Abundance of Probiotics of Infected Turtles

To further evaluate the improvement effect of purslane on the intestine of turtles, we compared the most abundant bacteria in the control, infection, and purslane groups, and screened the microbes that have been functionally identified as potential pathogens or probiotics, and analyzed the change in their relative abundance. The results showed that the abundance of the main potential pathogens (*Citrobacter freundii*, *Eimeria praecox*, and *Salmonella enterica*) was significantly increased after the challenge of *A. hydrophila* (*p* < 0.05, Figure 7). Interestingly, the abundance of all these potential pathogens was significantly reduced after purslane treatment (*p* < 0.05, Figure 7). For the probiotics, *A. hydrophila* significantly reduced the abundance of uncultured *Lactobacillus*, while purslane markedly elevated the abundance of uncultured *Lactobacillus* (*p* < 0.05, Figure 7).

## 3. Discussion

As a naturally widespread pathogen, *A. hydrophila* can infect a variety of aquatic animals [24]. In this study, Chinese pond turtles were challenged with *A. hydrophila-induced* symptoms containing ulceration of limbs, exposure of muscle, and a significant reduction in feeding and survival rates. By contrast, Chinese pond turtles in the control group showed a high feeding rate, 100% survival rate, and intact body surface. Similar results, including the wound healing of limb ulcers, and the significant improvement in feeding and survival rates, were also observed after immersion treatment with purslane. Nashwa et al. [25] also found that purslane significantly improved the resistance to *A. hydrophila* infection in Nile tilapia. Therefore, we speculate that purslane may have the antibacterial function and therefore improve the survival rate during *A. hydrophila* infection. Research on purslane has confirmed that it indeed has antibacterial and wound-healing properties [17,26], however, purslane contains multiple bioactive substances [27], and it is not clear from this study alone which component is responsible for healing decaying skin in Chinese pond turtles.

Intact intestinal barrier is necessary for animals to maintain nutrient absorption capacity [28], which is highly dependent on the length of the intestinal villus [29], the depth of the intestinal crypt [30], and the activity of digestive enzymes [31]. In this study, purslane significantly increased the villus height, crypt depth, and activity of intestinal digestive enzymes (α-amylase, lipase, and pepsin) in Chinese pond turtles infected with *A. hydrophila*. A study on Japanese quail found that purslane improved the digestive ability of quail by significantly increasing intestinal amylase and lipase activities [32], which is consistent with the results of this study. The increased digestive enzyme activities might be due to the antibacterial activity of purslane [33], lowering the risk of pathogens invading intestinal epithelial cells and promoting their ability to regenerate villi. The intestinal villus has the function of resistance to pathogenic bacterial infection [34], and the intestinal crypt is an invagination of the epithelium around the villus, which constantly divides to maintain the structure of the villus, and the increase in the depth of crypt could produce more developed intestinal villus [35]. Based on this, we speculate that purslane could enhance intestinal resistance to *A. hydrophila* by increasing intestinal villus height and crypt depth to improve intestinal morphology. 

The intestine, which is in direct contact with the external environment, may become a major site for colonization and growth of microbiota [36,37]. The intestinal microbiota plays an important role in maintaining the health of the host [38]. However, its composition and diversity were reported to be affected by the culture environment, including the water salinity, pH, feed source, and season [39,40,41,42]. In this study, for the microbial composition, it had been gradually changed with the infection, which might be one of the reasons causing the death of turtles. Thus, there should have a significant difference in intestinal microbial composition between the end-of-life turtles (infected turtles dead at day 11) and other groups’ turtles. As well, the comparison of the samples at day 11 (the end of lifespan in the infected turtles) to the samples at day 14 in other groups could clarify the effects of bacterial infection and purslane on turtles.

In this study, the number of OTUs and α-diversity of the intestinal microbiota of purslane was much higher than that in the infected group. Moreover, in combination with the β diversity analysis, it was speculated that purslane could increase the species abundance and community diversity of Chinese pond turtle intestinal microbiota. It had been reported that the Firmicutes and Bacteroidetes were the predominant phyla colonizing the healthy gut and played essential roles in host health-promoting and homeostasis [43]. In contrast, the phylum Proteobacteria is a microbial signature of dysbiosis in the intestinal microbiota [44]. In this study, we found that the intestinal microbiota was dominated by Firmicutes and Proteobacteria in the purslane group and the infected group, respectively, suggesting that purslane could improve the homeostasis of the intestinal microbiota and enhance the immunity of Chinese pond turtles [45].

In this study, we found a significant increase in the abundance of intestinal *Citrobacter freundii*, *Eimeria praecox* and *Salmonella enterica* in Chinese pond turtles infected with *A. hydrophila*. As a member of the genus *Citrobacter*, *Citrobacter freundii* has been noted to cause a variety of diseases including hemolysis, diarrhea, and enteritis [46,47]. It has been demonstrated that *Citrobacter freundii* can form drug-resistant biofilms to resist antibiotics [48], and Soliman et al. [49] found that purslane can suppress the growth of *Candida albicans* by inhibiting its biofilm formation, so, we conclude that the reduction in the abundance of *Citrobacter freundii* found in this study may be related to the inhibition of its biofilm formation and growth by purslane. *Salmonella enterica* is a member of the *Salmonella* spp., as sentinels of *Salmonella* spp. invasion, aquatic turtles are highly susceptible to accumulating Salmonella present in water and cause their disease [50]. Raidal et al. [51] also found that *salmonellae* can cause systemic illness and death in green sea turtles. *Eimeria praecox* can infect the duodenum of broiler chickens and affect their growth, which in turn shows pathogenicity [52]. Therefore, in this study, we found that purslane significantly reduces the abundance of intestinal *Salmonella enterica* and *Eimeria praecox*, thus protecting the intestinal health of turtles, but the exact mechanism of its reduction of *Salmonella enterica* and *Eimeria praecox* remains to be further investigated. In addition, uncultured *Lactobacillus* was also noted to be significantly lower in abundance in the *A. hydrophila* infection group. Although uncultured *Lactobacillus* is not annotated as any species, *Lactobacillus* spp. is considered a probiotic because of its use in maintaining the activity of the normal microbiota in humans and other mammals [53]. Studies on *Cyprinus carpio* Huanghe var [54], crucian carp [55], hybrid catfish [56] and Nile tilapia [57] found that *Lactobacillus* improved their immunity to *A. hydrophila*. In this study, purslane increased the abundance of intestinal *Lactobacillus* in Chinese pond turtles, and the same results were found in broilers [58] and mice [21], suggesting that purslane may improve the intestinal immunity of turtles by increasing the abundance of *Lactobacillus* to resist *A. hydrophila* infection.

## 4. Materials and Methods

### 4.1. Experimental Turtle and Design

A total of 120 healthy juvenile Chinese pond turtles (5.1 ± 0.5 cm, 32.4 ± 4.7 g) were purchased from Gu Shangxiang Farm (Chengdu, China), and randomly assigned to rectangular glass 120 × 60 × 80 cm^3^ tanks for at least one week of acclimatization. After acclimatization, turtles were randomly divided into three groups, each group includes three tanks, 10 turtles per tank. The first group was the control group, where the turtles were bred in clean water. Turtles in the second group (*A. hydrophila* infection group) were challenged with *A. hydrophila* (1 × 10^5^ CFU/mL) by immersion method. The third group was the infection plus purslane group, in which turtles were treated with purslane under the immersion of *A. hydrophila* (1 × 10^5^ CFU/mL) suspension, the immersion was performed by adding purslane powder (purity ≥ 99%) to *A. hydrophila* suspension and adjusting the purslane concentration to 6.25 g/L. The immersion was carried out over the turtle bodies and sustained until the end. During the experiment, the water temperature, dissolved oxygen, pH, ammonia nitrogen, and nitrite were maintained at 18.3 ± 1.6 °C, 7.8 ± 0.4 mg/L, 6.5 ± 0.5, ≤0.01 mg/L, and ≤0.05 mg/L, respectively. The breeding water was renewed once every two days, and the volume of water renewal is 50% of the total volume.

The turtles were fed with commercial feed (Haida Group Co., Ltd., Guangzhou, China) two times per day (at 8:00 a.m. and 20:00 p.m., respectively), each fed 3% of the total turtle’s weight. 1 h after, the residual feed was collected and weighed after drying to calculate the feeding rate (average of the twice feeding rate per day). From the first day of the breeding, the cumulative survival rate of each group was counted daily, cumulative survival rate (%) = (currently surviving turtles/total turtles) × 100. The *A. hydrophila* strain was friendly provided by the laboratory of the Department of Fisheries, College of Animal Science and Technology, Sichuan Agricultural University (Chengdu, China), and 1 × 10^5^ CFU/mL was the lowest concentration found to cause high mortality in turtles based on our preliminary experiment. The purslane powder used in this experiment was obtained from Sichuan Hrtd Biological Technology Co., Ltd. (Chengdu, China), and 6.25 g/L purslane solution is the IC_50_ concentration found to inhibit the growth of *A. hydrophila* in the pre-experiment in vitro. All animal handling procedures were approved by the Animal Care and Use Committee of Sichuan Agricultural University, following the guidelines of animal experiments of Sichuan Agricultural University under permit number 035-2212129116.

### 4.2. Sample Collection 

During breeding, abnormal changes in turtle skin (e.g., erosions and ulcers) were recorded with reference to the description of Muñoz et al. [59]. After two weeks of administration, eight survival turtles were randomly selected from each group and anesthetized with 10% diethyl ether (Jinjiang Aquatic Supplies Co., Ltd., Fujian, China), and their intestines were collected and stored at −80 °C until for use. Because the death of infected turtles happened at day 8, and most of the administrated turtles were dead on day 11, we increased the check frequency to find the new dead turtles and tried to collect intestines when the turtles were dead in 1 h in the infection group. Among these intestinal samples, four were used for microbial diversity analysis (n = 4), while the rest samples were used for digestive enzyme activity assays (n = 4).

### 4.3. Histological Observation

Intestinal samples were collected and then preserved in 10% neutral buffered formalin for at least 24 h for fixation. The fixed intestinal samples were dehydrated, transparentized with xylene, and embedded in paraffin wax. The solidified wax blocks were cut into 5-mm slices and mounted on slides for H&E staining. After staining, the slides were observed under an optical microscope (Nikon, Tokyo, Japan). Villus height and crypt depth were observed and recorded, the image was processed and analyzed using ImageJ software (version 1). To study the effects of purslane on the intestine during *A. hydrophila* infection, we evaluated the intestinal damage based on a scoring system proposed by Baums et al. [60]. Briefly, abnormal changes in the mucosa of the intestine, such as hyperplasia, necrosis, deposits, hypertrophy, or atrophy, were assessed using a score of 0–5: (0) unchanged; (1) mild altered; (2) general altered (3) moderate altered; (4) large altered and (5) severe altered (diffuse necrosis).

### 4.4. Digestive Enzyme Activity Assays of Intestine

The intestines were ground with buffer solution using commercial kits (A080-1-1, C016-1-1, A054-1-1, Nanjing Jiancheng Bioengineering Institute, Nanjing, China). The activity of pepsin, α-amylase, and lipase was assayed with the above-mentioned commercial kits. All operations were performed in strict accordance with the manufacturer’s description.

### 4.5. DNA Isolation, Library Construction, and Sequencing

Genomic DNA was extracted from the intestinal bacteria using a bacterial DNA isolation kit (Foregene, Chengdu, China) according to the manufacturer’s instructions. The extracted DNA was checked on 1% agarose gel, and DNA concentration and purity were determined with NanoDrop 2000 UV-vis spectrophotometer (Thermo Scientific, Wilmington, DE, USA). After the extraction, DNA fragments from the samples were amplified using specific primers: 338F: 5′-ACTCCTACGGGAGGCAGCAG-3′, 806R: 5′-GGACTACHVGGGTWTCTAAT-3′ [61]. The PCR product was detected using 2% agarose gel, purified using the AxyPrep DNA Gel Extraction Kit (Axygen Biosciences, Union City, CA, USA) and quantified using Quantus™ Fluorometer (Promega, Madison, WI, USA). V3–V4 amplicon library was constructed using TruSeqTM DNA Sample Prep Kit (Illumina, San Diego, CA, USA). High-throughput sequencing was performed in a paired-end model using the Illumina MiSeq PE300 platform (Illumina, San Diego, CA, USA) by Majorbio Bio-Pharm Technology Co., Ltd. (Shanghai, China).

### 4.6. Taxonomy Classification and Differential Abundance Analysis

After MiSeq sequencing, based on the overlapping relationship between PE reads, pairs of reads were merged into a sequence using Flash software (version 1.2.11). Raw fastq files were demultiplexed, quality-filtered with the following criteria: (i) the 300 bp reads were truncated at any site receiving an average quality score <20 in a 50 bp sliding window, discarding the truncated reads <50 bp, (ii) splicing pairs of reads into a sequence based on the overlapping relationship between PE reads, assembling only sequences with an overlap of more than 10 bp, (iii) the maximum mismatch ratio allowed in the overlap region of a spliced sequence was 0.2, after removing non-conforming sequences. The acquired sequences were clustered into operational taxonomic units (OTUs) at a 70% confidence level by Uparse (version 11, http://drive5.com/uparse (accessed on 12 May 2022)) via the SILVA rRNA database. The RDP Classifier was used to obtain OTU annotation information at a 97% similarity level. Alpha diversity, including Shannon, Simpson, Chao and Ace index were calculated by Mothur (version 1.30.2). In addition, beta diversity and hierarchical clustering trees were constructed by the unweighted pair group method with arithmetic mean (UPGMA) were used to compare the similarity of comparative cluster composition. Hierarchical clustering trees were constructed by IQ-TREE (version 1.6.8, http://www.iqtree.org/ (accessed on 18 May 2022)), PCA statistical analysis and graphing with The R Programming Language (version 3.3.1, http://www.r-project.org/ (accessed on 18 May 2022)). Species with relative abundance less than 0.01 in all samples were categorized as “other”. 

### 4.7. Statistical Analysis

Data were expressed as mean ± SD. Statistical analysis was performed using one-way ANOVA in SPSS version 27.0 software. Least Significant Difference (LSD) was used to analyze the differences among the groups. Principal coordinates analysis was performed to cluster different samples according to explanatory parameters. A *p*-value less than 0.05 was considered as the statistical significance.

## 5. Conclusions

Our study provided direct evidence that purslane improves the intestinal health of Chinese pond turtles infected with *A. hydrophila*. The purslane group presented with the repair of limb epidermis and intestinal morphology, higher survival and feeding rates, and increased digestive enzyme activity after *A. hydrophila* infection. The function of purslane in inhibiting the proportion of potential pathogens (*Citrobacter freundii*, *Salmonella enterica*, and *Eimeria praecox*), increasing the abundance of probiotic (uncultured *Lactobacillus*), and protecting the intestinal microbial barrier was also revealed. The information presented in this paper will provide a theoretical strategy for antimicrobial studies in wild and farmed turtles and expand the application of purslanes in aquaculture.

## Figures and Tables

**Figure 1 ijms-24-10260-f001:**
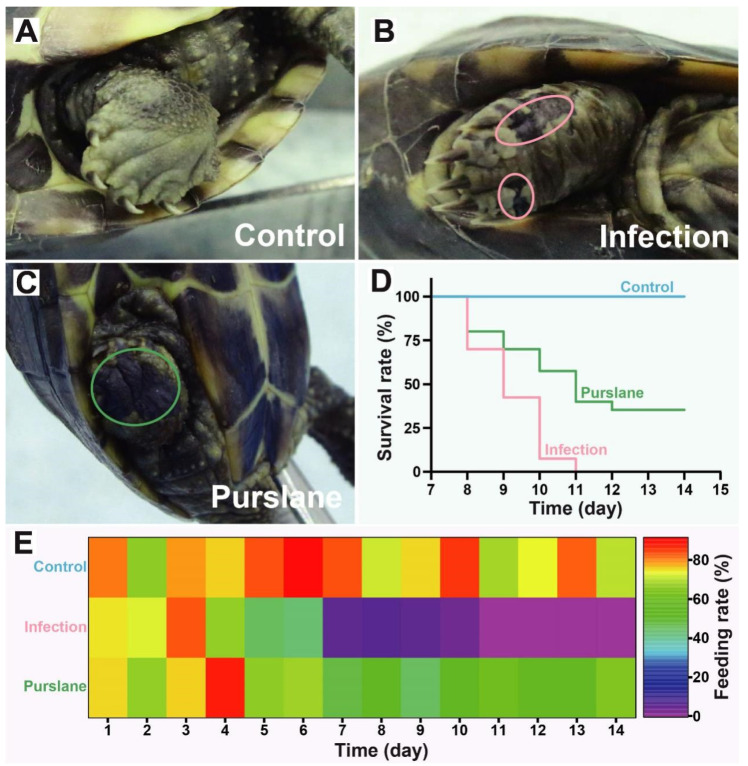
Effect of purslane on symptoms of the body surface, survival, and feeding rates of Chinese pond turtles during *A. hydrophila* infection. (**A**–**C**) Example of symptoms on the body surface of the limbs. The pink circle represents the exposed muscle layer and epidermal ulceration, the green circle indicates the neonatal black epidermis. (**D**) The cumulative survival rate of each group. Survival rate (%) = (currently surviving turtles/total turtles) × 100. (**E**) Heat map of the feeding rate of each group. Feeding rate (%) = (1 − average residual feed/average feeding amount) × 100.

**Figure 2 ijms-24-10260-f002:**
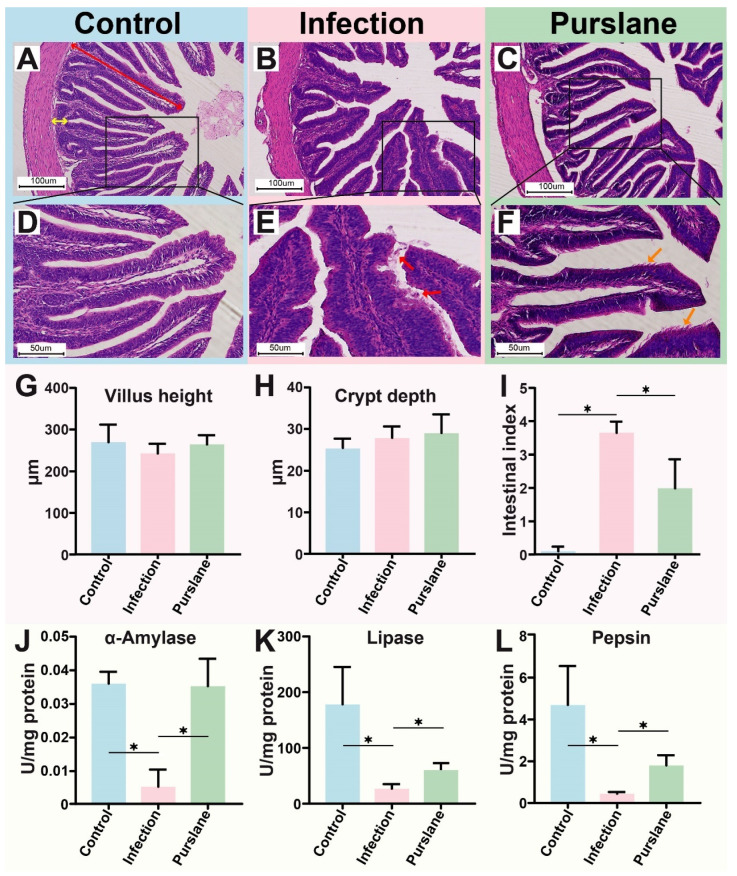
Effect of purslane on intestinal structure and digestive enzyme activities of Chinese pond turtles during *A. hydrophila* infection. (**A**–**F**) Histopathological changes in the control (**A**,**D**), infection (**B**,**E**) and purslane (**C**,**F**) groups of Chinese pond turtle’s intestine. The red bidirectional arrow shows villus height and the yellow bidirectional arrow indicates crypt depth. Red unidirectional arrow means shed intestinal mucosa and the orange unidirectional arrow represents newly generated intestinal mucosa. (**G**,**H**) Villus height and crypt depth in different groups. (**I**) Intestinal health status (organ index) of Chinese pond turtles in different groups. (**J**–**L**) The activity of α-Amylase, lipase, and pepsin in the Chinese pond turtle’s intestine. One-way ANOVA plus Bonferroni post-tests, * *p* < 0.05.

**Figure 3 ijms-24-10260-f003:**
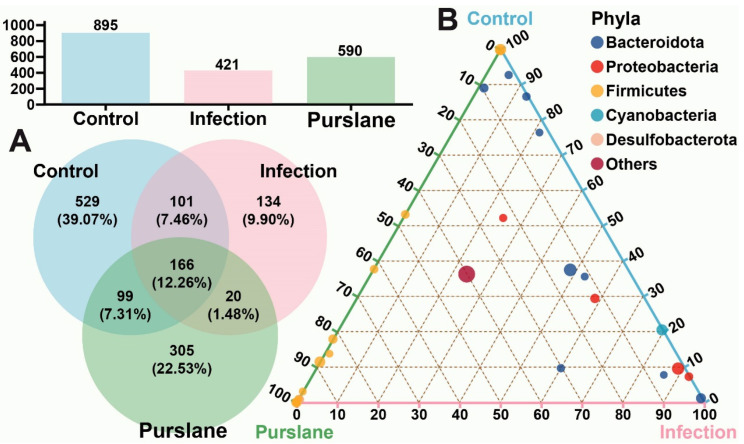
OTU differential analysis of control, infection, and purslane groups. (**A**) Histogram (above) and Venn diagram (below) of OTUs number in different groups. (**B**) OTU ternary analysis of different groups in the Chinese pond turtle intestines. The three corners represent the control, infection, and purslane samples, and the color of solid circles in the figure represents the annotation of OTUs at the phyla level, and the size of the circles stands for the average relative abundance (%) at the phyla level.

**Figure 4 ijms-24-10260-f004:**
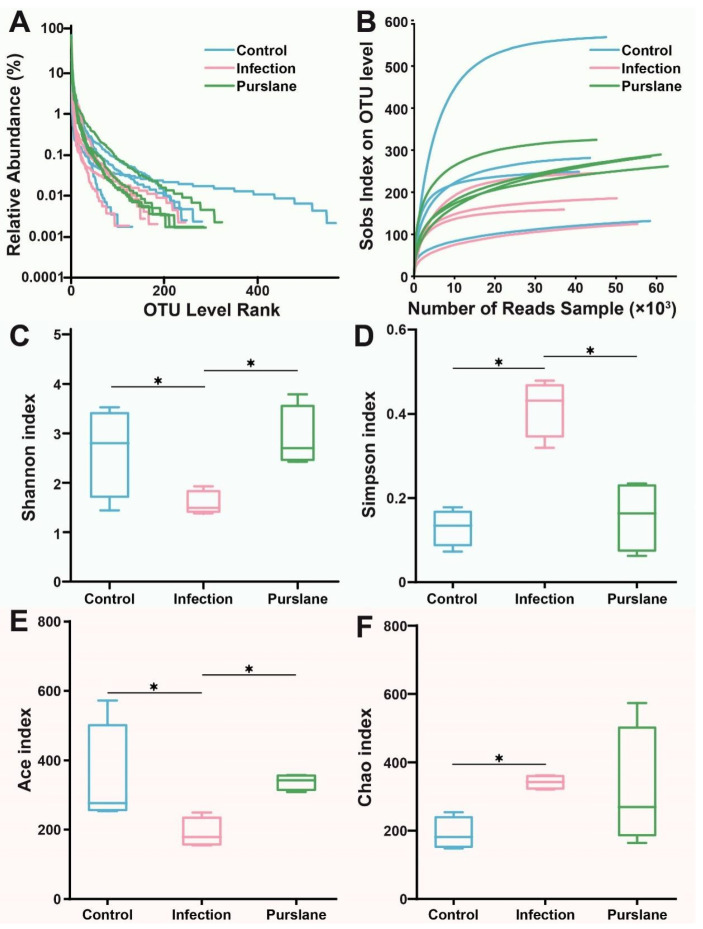
Species annotation and evaluation. (**A**) Rank-Abundance curve at the OTU level. (**B**) Sobs index at the OTU level. (**C**,**D**) Shannon and Simpson index at the OTU level. (**E**,**F**) Ace and Chao index at the OTU level. * *p* < 0.05 indicates a significant difference between the two groups, one-way ANOVA plus Bonferroni post-tests.

**Figure 5 ijms-24-10260-f005:**
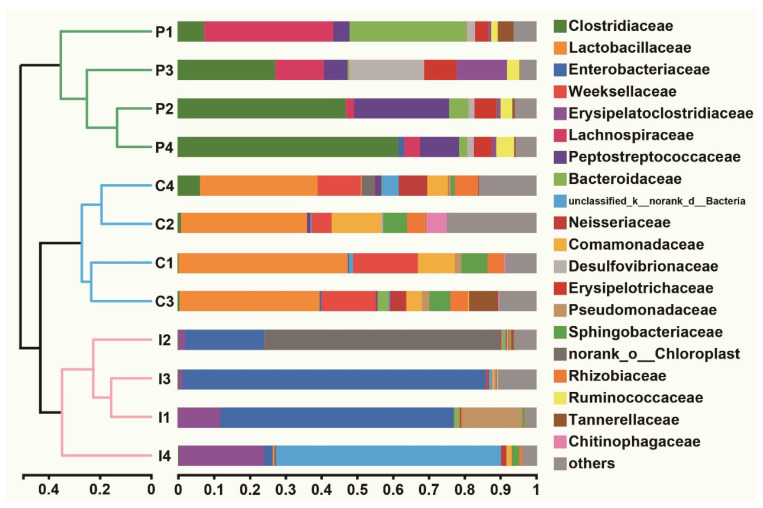
Hierarchical clustering trees of Chinese pond turtle intestines at the family level. Clustering analysis based on Bray–Curtis dissimilarity.

**Figure 6 ijms-24-10260-f006:**
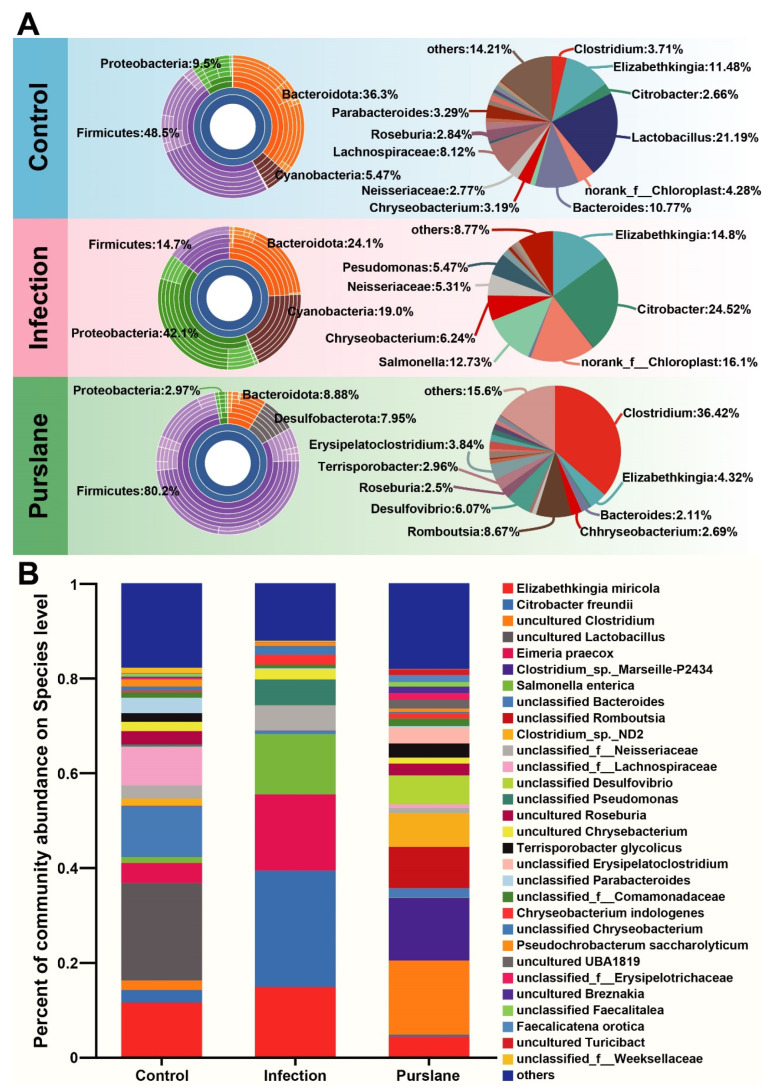
Comparison of microbial community structures among control, infection, and purslane groups. (**A**) Sunburst plots at the phyla level and pie charts of the most abundant genus. (**B**) Species composition of each sample.

**Figure 7 ijms-24-10260-f007:**
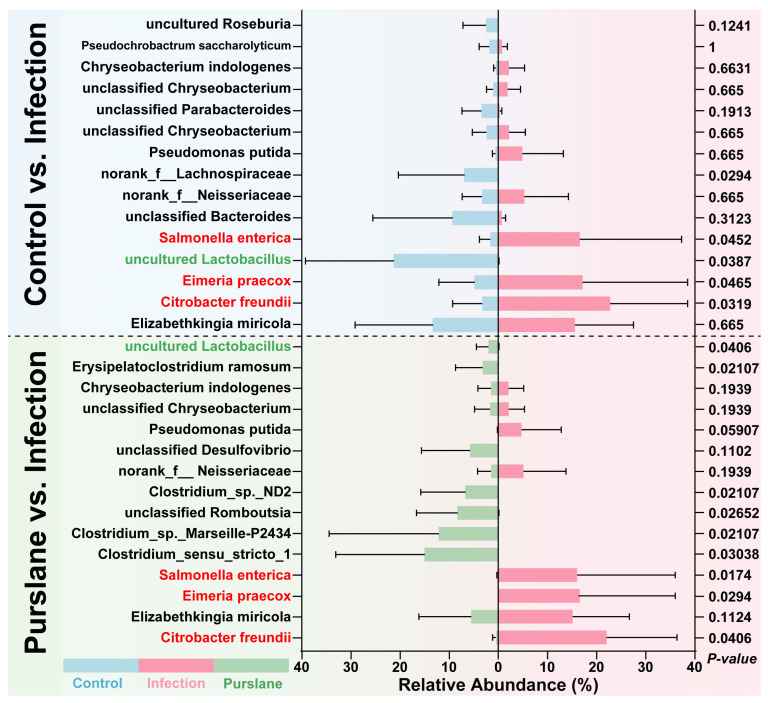
Comparison of microbial community differences between control or purslane and Infection groups. The chart lists the top 15 abundant species in control, infection, and purslane samples, and data are presented as means ± standard deviation. Wilcoxon rank sum test was applied to determine if differences between groups were significant (*p* < 0.05). The communities with red and green fonts indicate potential pathogens and probiotics with significant changes in abundance, respectively.

**Table 1 ijms-24-10260-t001:** Raw data statistics of sequencing samples.

Sample Category	Replicate	Seq_Num	Base_Num	Mean_Length	Min_Length	Max_Length
Control	C1	42,201	17,744,244	420.46	208	528
	C2	46,657	19,623,401	420.58	254	486
	C3	44,444	18,657,768	419.80	205	486
	C4	48,767	20,723,904	424.95	208	458
Infection	I1	46,145	18,921,528	410.04	247	490
	I2	51,282	21,729,312	423.72	208	499
	I3	37,434	14,148,899	377.96	201	525
	I4	58,329	24,967,742	428.05	221	486
Purslane	P1	65,251	26,607,757	407.77	240	486
	P2	73,938	30,679,329	414.93	227	433
	P3	63,494	26,246,140	413.36	219	486
	P4	73,640	30,037,311	407.89	214	433

## Data Availability

The raw 16S rRNA sequencing data have been submitted to the Sequence Read Archive (SRA) database of the National Center for Biotechnology Information (NCBI), with accession number PRJNA957584.

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
