# Peer review of "Effect of Purslane (*Portulaca oleracea* L.) on Intestinal Morphology, Digestion Activity and Microbiome of Chinese Pond Turtle (*Mauremys reevesii*) during *Aeromonas hydrophila* Infection"

_ijms, 2023, doi:10.3390/ijms241210260_

Round 1
Reviewer 1 Report
The article's topic is interesting. The A. hydrophila impact on pond turtles M. reevesii is well-presented and the investigation was planned correctly.
The small mistakes I spotted in the text do not have big influence on the scientific merit of the article.

Author Response
Point-by-point response to Reviewer1
Dear Reviewer:
On behalf of my co-authors, we thank you very much for giving us an opportunity to revise our manuscript, we appreciate a lot for your positive and constructive comments and suggestions on our manuscript.
We have amended the mistakes in the manuscript one by one as you suggested, and the specific corrections are shown in red in the Line 24, Line 32, Line 82, Line 84, Line 183, Line 277 and Line 279 of the manuscript.

Reviewer 2 Report
The manuscript entitled “Effect of purslane on intestinal morphology, digestion activity and microbiome of Chinese pond turtle (Mauremys reevesii) during Aeromonas hydrophila infection” by Yang et al has demonstrated the effectiveness of purslane towards the repairment of limb epidermis and intestine. The results are interesting and presented well. The manuscript can be published with some minor changes:
1. Author can elaborate in the method section how they have soaked the turtles with purslane for the infection group.
2. Author should discuss the plausible mechanism for purslane’s effectiveness.
Author Response
Point-by-point response to Reviewer 2
Dear Reviewer:
Thank you very much for your advice on our manuscript. We have considered your comments carefully and have made revision in the paper.
Comment 1: Author can elaborate in the method section how they have soaked the turtles with purslane for the infection group.
Respond 1: Thanks for your valuable comments to improve the quality of our manuscript. The immersion method has been described in more detail as you suggested, and the sentence " The third group was infection plus purslane group, in which turtles were treated with purslane under the immersion of Aeromonas hydrophila (1×105 CFU/mL) suspension, the immersion was performed by adding purslane powder (purity ≥99%) to Aeromonas hydrophila suspension and adjusting the purslane concentration to 6.25 g/L. The immersion was carried out to over the turtle bodies and sustained until to the end" has been added to the manuscript (Line 295-300).
Comment 2. Author should discuss the plausible mechanism for purslane’s effectiveness.
Respond 2: Thank you for your advice, we have discussed the possible mechanism for purslane’s effectiveness in revision version. The specific discussions are as follows:
Line 236-238: Based on this, we speculate that purslane could enhance intestinal resistance to A. hydrophila by increasing intestinal villus height and crypt depth to improve intestinal morphology.
Line 257-260: In this study, we found that the intestinal microbiota was dominated by Firmicutes and Proteobacteria in the purslane group and the infected group, respectively, suggesting that purslane could improve the homeostasis of the intestinal microbiota and enhance immunity of Chinese pond turtles.
Line 267-269: so, we conclude that the reduction in abundance of Citrobacter freundii found in this study may be related to the inhibition of its biofilm formation and growth by purslane.
Line 283-286: In this study, purslane increased the abundance of intestinal Lactobacillus in Chinese pond turtle, and the same results were found in broilers and mice, suggesting that purslane may improve the intestinal immunity of turtle by increasing the abundance of Lactobacillus to resist A. hydrophila infection.

Reviewer 3 Report
The manuscript by S. Yang et al. entitled "Effect of purslane on intestinal morphology, digestion activity and microbiome of Chinese pond turtle (Mauremys reevesii) during Aeromonas hydrophila infection" is dovoted to the activity of purslane (Portulaca oleracea L.) on Chinese pond turtle during A. hydrophila infection. The authors used microbiome sequencing.
The manuscript is of some interest. However, there are the following shortcomings and suggestions.
Title: I suggest adding (Portulaca oleracea L.) after purslane.
Throughout the manuscript: Carefully check that all Latin names are italicized throughout the manuscript! For example, lines 3, 4, 23, 24, 27, 130, etc. Also, shorten the species to one letter and a dot if it appears a second or more times in the text.
Line 32: Capitalize S for Salmonella.
Line 37: These appear to be conditions, not symptoms. Symptoms include fever, cough, etc.
Line 74: Perhaps symptoms?
Line 97: What is H&E?
Lines 140, 165, 166: Typos: OUT
Figure 4: What does "*" mean?
Figure 4: I would like to know the authors' opinion as to why control (C) and infection (I) are clustered together.
Figure 7: It is not clear what the difference is between the upper and lower halves divided by the dashed line.
Line 208: Replace with "feeding and survival rates".
Line 211: Do not italicize Nile tilapia.
Line 262: Do not italicize Salmonella.
Line 269, 271: Do not italicize "incultured".
Line 324: Add descriptions for grades (2) and (4).
Line 338: Where did these primers come from? Reference?
Line 342: Add details for Illumina sequencing, e.g. which kit was used for library preparation, insert size, etc?
Line 345: Capitalize S in MiSeq
Line 351: What software was used to construct trees?
Line 358: What software was used for PCA?
Since the authors are not native English speakers, I would recommend a better polishing of the sentences in the Introduction. Sometimes it is not clear what the authors wanted to say.
Author Response
Point-by-point response to the reviewer 3
Dear Reviewer:
On behalf of my co-authors, we thank you very much for giving us an opportunity to revise our manuscript. Your comments are all valuable and very helpful for revising and improving our paper, as well as the important guiding significance to our researches.
The manuscript by S. Yang et al. entitled "Effect of purslane on intestinal morphology, digestion activity and microbiome of Chinese pond turtle (Mauremys reevesii) during Aeromonas hydrophila infection" is devoted to the activity of purslane (Portulaca oleracea L.) on Chinese pond turtle during A. hydrophila infection. The authors used microbiome sequencing. The manuscript is of some interest. However, there are the following shortcomings and suggestions.
Comment 1: Title: I suggest adding (Portulaca oleracea L.) after purslane.
In response: We have added Portulaca oleracea L. to the title as you suggested. (Line 2)
Comment 2: Throughout the manuscript: Carefully check that all Latin names are italicized throughout the manuscript! For example, lines 3, 4, 23, 24, 27, 130, etc. Also, shorten the species to one letter and a dot if it appears a second or more times in the text.
In response: Thank you for your careful review. We have carefully revised the manuscript as you suggested, and have repeatedly checked the manuscript to ensure that similar mistakes do not occur in the manuscript (Line 24, Line 183, Line 199, Line 200, Line 277 and Line 279). The second or more times species appear in the text that require abbreviations have also been modified as you suggested (Line 13, Line 16, Line 18, Line 22, Line 26, Line 36, etc.).
Comment 3: Line 32: Capitalize S for Salmonella.
In response: Thank you for your careful review. We have modified the error as you suggested. (Line 24)
Comment 4: Line 37: These appear to be conditions, not symptoms. Symptoms include fever, cough, etc.
In response: Thanks for your suggestion, we have modified " symptoms " to " conditions " as you suggested. (Line 38)
Comment 5: Line 74: Perhaps symptoms?
In response: Yes, it should be symptoms here. (Line 75)
Comment 6: Line 97: What is H&E?
In response: H&E stain stands for " Hematoxylin and eosin stain ". We have added the descriptions in 2.2 chapter section. (Line 98)
Comment 7: Lines 140, 165, 166: Typos: OUT
In response: Thank you for your careful review. We have modified "OUT" to "OTU" as you suggested. (Line 140, Line 141, Line 167, Line 168)
Comment 8: Figure 4: What does "*" mean?
In response: "*" represents a significant difference between the two groups, i.e., p<0.05, and we have inserted the corresponding description in the figure notes of Figure 4. (Line 168-169)
Comment 9: Figure 4: I would like to know the authors' opinion as to why control (C) and infection (I) are clustered together.
In response: Thanks for your question. The cluster analysis showed that the control (C) and infection (I) groups should just have some samples clustered together, but overall, they were still separate. This may be due to the similarity of microbial communities between samples of different treatment groups. And there are individual differences between samples of the same treatment. (Line 171, Figure 5)
Comment 10: Figure 7: It is not clear what the difference is between the upper and lower halves divided by the dashed line.
In response: Thanks for your comment. The upper and lower halves of the dashed lines in Figure 7 represent " infected group vs control group " and " purslane group vs control group ", respectively. In Figure 7, we have added labels to make reader to clearly recognize the result representation. (Line 201)
Comment 11: Line 208: Replace with "feeding and survival rates".
In response: Thanks for your careful review. We have replaced " feeding rate and survival rate " with " feeding and survival rates" as you suggested. (Line 212, Line 214)
Comment 12: Line 211: Do not italicize Nile tilapia.
In response: We have revised this mistake as you suggested. (Line 217)
Comment 13: Line 262: Do not italicize Salmonella.
In response: We have revised this mistake as you suggested. (Line 271)
Comment 14: Line 269, 271: Do not italicize " uncultured ".
In response: We have revised this mistake as you suggested. (Line 277, Line 279)
Comment 15: Line 324: Add descriptions for grades (2) and (4).
In response: According to your suggestion, the descriptions of the grades " (2) general altered " and " (4) large altered " were added. (Line 341-342)
Comment 16: Line 338: Where did these primers come from? Reference?
In response: These primers are derived from existing literature, and we have added reference to these primers in the manuscript as you suggested in Line 356. https://www.nature.com/articles/s41467-021-27857-6
Comment 17: Line 342: Add details for Illumina sequencing, e.g., which kit was used for library preparation, insert size, etc?
In response: Illumina sequencing-related details, such as library construction, data quality control, etc., have been added to the materials and methods as you suggested. (Line 358-359, Line 364-371)
Comment 18: Line 345: Capitalize S in MiSeq
In response: We have amended this mistake as you suggested. (Line 364)
Comment 19: Line 351: What software was used to construct trees?
In response: IQ-TREE software (version 1.6.8) was used to construct the hierarchical clustering tree, which we have added to the description in the Materials and Methods section as you suggested. (Line 378-379)
Comment 20: Line 358: What software was used for PCA?
In response: The R Programming Language (version 3.3.1) was used for PCA statistical analysis and graphing, and we have provided additional descriptions in the Materials and Methods section as you suggested. (Line 379-381)
Comment 21: Since the authors are not native English speakers, I would recommend a better polishing of the sentences in the Introduction. Sometimes it is not clear what the authors wanted to say.
In response: Thanks for your suggestion. In order to improve the quality of the manuscript, we have requested a native English speaker to correct the English writing.

Reviewer 4 Report
Manuscript ID ijms-2421065
GENERAL REMARKS
Dear authors,
I evaluated the manuscript entitled "Effect of purslane on intestinal morphology, digestion activity and microbiome of Chinese pond turtle (Mauremys reevesii) during Aeromonas hydrophila infection". I consider your research interesting and, at the same time, the manuscript well-constructed: the experimental design is appropriate, the results adequately presented and discussed, and the conclusions well defined. I cannot say the same about the description of the materials and methods, which in my opinion need to be reviewed in several parts. Also, the introduction needs to be revised on some points. My specific comments are listed below, point by point. I hope my observations will lead to improvements in the manuscript.
SPECIFIC COMMENTS
L 31: the first reference ([1]) should not appear as an index. Please check, thanks.
L 34: I find it more correct that "hindrance" is declined in the plural. Thanks.
L 65-66: I find it more correct that the reference number (i.e., [23]) directly follows the reference it refers to (Zhang et al.). Furthermore, it would also be useful to indicate which species the cited study refers to. Thanks.
L 69-72: In my opinion, the adjective "unclear" suggests that the use of purslane to protect the turtles' intestines against the infectious action of Aeromonas hydrophila has already been studied. If so, the authors should indicate the studies that have addressed this issue and explain in this regard the state of the art with respect to which the research in question fits. Conversely (as I believe), if there are no studies in this regard, the adjective "unclear" should be replaced with "unknown", for example. Thanks.
L 205-208: To give greater emphasis to the results obtained, it could be useful, in my opinion, to also recall the average mortality rate of turtles under ordinary farming conditions, as well as the usual feeding rate under feeding conditions like those under which the study was conducted. Thanks.
L 288-290: I apologize for the pedantry, but I think it could be useful, above all for the purposes of the replicability of the experiment, on what the choice of the level of CFU used in the experiment was based. For the same reason (replicability), I invite the authors to describe, albeit briefly, the immersion method. Moreover, if the turtles have been "soaked" with purslane I assume it is a purslane-based liquid (extract). However, this is not explicit. I invite the authors to implement this part, providing more details on the production technique of the "liquid purslane" used for the experimental purpose. Thanks.
L 295: I invite the authors to provide details of the type of experimental diet used, if possible. Thanks.
L 297: In my opinion, adding a period would improve the sentence (for example, after "turtle's weight"). Thanks.
L 301-302: I do not understand which preliminary experiment the authors refer to. What purpose do they have? Can the authors provide further details? Thanks.
L 307: I would also add the mortality rate and the related evaluation procedure among the materials and methods. Thanks.
L307-312: I do not know the description of the symptoms and signs of the body surface produced by the infection (e.g., ulcers, black epidermis, etc.) discussed in the results (L 76-87) nor even more of the relative evaluation criteria. Please provide these details, thanks.
L 308: It is not clear to me whether the group from which the animals were selected is intended as a treatment or replication (i.e., the three tanks cited in line 286) within each treatment. Thanks.
Author Response
Point-by-point response to reviewer 4
Dear Reviewer:
On behalf of my co-authors, we thank you very much for giving us an opportunity to revise our manuscript. Your comments are all valuable and very helpful for revising and improving our paper, as well as the important guiding significance to our researches.
Comment 1: I evaluated the manuscript entitled "Effect of purslane on intestinal morphology, digestion activity and microbiome of Chinese pond turtle (Mauremys reevesii) during Aeromonas hydrophila infection". I consider your research interesting and, at the same time, the manuscript well-constructed: the experimental design is appropriate, the results adequately presented and discussed, and the conclusions well defined. I cannot say the same about the description of the materials and methods, which in my opinion need to be reviewed in several parts. Also, the introduction needs to be revised on some points. My specific comments are listed below, point by point. I hope my observations will lead to improvements in the manuscript.
In response: Sincerely thank you for your reviews. We have carefully considered your comments and revised the corresponding parts of the manuscript line by line according to your suggestions, with red color in the revised manuscript.
Comment 2: L 31: the first reference ([1]) should not appear as an index. Please check, thanks.
In response: Thank you for your careful review. We have corrected this mistake as you suggested by cancelling the index of the first reference ([1]). (Line32)
Comment 3: L 34: I find it more correct that "hindrance" is declined in the plural. Thanks.
In response: Thanks for your valuable suggestion, we have rewritten " hindrance " to " hindrances" as you suggested. (Line35)
Comment 4: L 65-66: I find it more correct that the reference number (i.e., [23]) directly follows the reference it refers to (Zhang et al.). Furthermore, it would also be useful to indicate which species the cited study refers to. Thanks.
In response: Thanks for your valuable suggestion. We have revised the placement of the reference numbers as you suggested (directly follows the reference it refers to), and have double-checked the full manuscript to ensure that there are no more similar mistakes, and have indicated the study species of the cited reference – mice. (Line 66-67)
Comment 5: L 69-72: In my opinion, the adjective "unclear" suggests that the use of purslane to protect the turtles' intestines against the infectious action of Aeromonas hydrophila has already been studied. If so, the authors should indicate the studies that have addressed this issue and explain in this regard the state of the art with respect to which the research in question fits. Conversely (as I believe), if there are no studies in this regard, the adjective "unclear" should be replaced with "unknown", for example. Thanks.
In response: Many thanks for your careful review, we have modified " unclear " to " unknown " as you suggested. (Line 70)
Comment 6: L 205-208: To give greater emphasis to the results obtained, it could be useful, in my opinion, to also recall the average mortality rate of turtles under ordinary farming conditions, as well as the usual feeding rate under feeding conditions like those under which the study was conducted. Thanks.
In response: Thanks for your valuable suggestion. The mortality rate and feeding rate of turtles under ordinary farming conditions have been added as you suggested. (Line 212-213)
Comment 7: L 288-290: I apologize for the pedantry, but I think it could be useful, above all for the purposes of the replicability of the experiment, on what the choice of the level of CFU used in the experiment was based. For the same reason (replicability), I invite the authors to describe, albeit briefly, the immersion method. Moreover, if the turtles have been "soaked" with purslane I assume it is a purslane-based liquid (extract). However, this is not explicit. I invite the authors to implement this part, providing more details on the production technique of the "liquid purslane" used for the experimental purpose. Thanks.
In response: Thanks for your valuable comments to improve the quality of our manuscript. The 1×105 CFU/mL Aeromonas hydrophila used in this study was the lowest concentration found to cause mortality in turtles based on our preliminary experiment (results not presented), and we have added an explanation for the choice of this concentration in the materials and methods section as you suggested (Line 311-313). The immersion method has also been described in more detail as you suggested, and the sentence " The third group was infection plus purslane group, in which turtles were treated with purslane under the immersion of A. hydrophila (1×105 CFU/mL) suspension, the immersion was performed by adding purslane powder (purity ≥ 99%) to A. hydrophila suspension and adjusting the purslane concentration to 6.25 g/L. The immersion was carried out to over the turtle bodies and sustained until to the end." has been added to the manuscript (Line 295-300). The purslane used in this experiment is a finished product that we purchased directly from Sichuan Hrtd Biological Technology Co., Ltd, and according to the instruction, it is a brown powder with a purity ≥99%. In this study, we prepared the purslane powder as a 6.25 mg/L solution for use. This concentration was chosen because pre-experiments found it to be the lowest concentration that inhibited the growth of Aeromonas hydrophila. And we have added a description of purslane in the manuscript as you suggested. (Line 313-316)
Comment 8: L 295: I invite the authors to provide details of the type of experimental diet used, if possible. Thanks.
In response: Thanks for your question. The turtle diet used in this experiment is a commercial feed that we purchased from Haida Group. We are sorry that the company is not willing to disclose the specific ingredients of the feed to us due to copyright reason, and we hope that our explanation will be understood by you.
Comment 9: L 297: In my opinion, adding a period would improve the sentence (for example, after "turtle's weight"). Thanks.
In response: Thanks for your suggestion, we have modified "," to "." as you suggested. (Line 306)
Comment 10: L 301-302: I do not understand which preliminary experiment the authors refer to. What purpose do they have? Can the authors provide further details? Thanks.
In response: Thanks for your question. The preliminary experiments mentioned in this manuscript were aimed at finding the optimum concentration of Aeromonas hydrophila and purslane solution in this study, we apologize for the perplexity, it was a writing error and we have added a description of the preliminary experiments in the Materials and Methods. (Line 311-316)
Comment 11: L 307: I would also add the mortality rate and the related evaluation procedure among the materials and methods. Thanks.
In response: Thanks for your valuable suggestion. The evaluation of the survival rate has been described in the materials and methods as you suggested. (Line 307-309)
Comment 12: L307-312: I do not know the description of the symptoms and signs of the body surface produced by the infection (e.g., ulcers, black epidermis, etc.) discussed in the results (L 76-87) nor even more of the relative evaluation criteria. Please provide these details, thanks.
In response: Thanks for your valuable suggestion. We have added evaluation criteria for these symptoms as you suggested. (Line 321-322)
Comment 13: L308: It is not clear to me whether the group from which the animals were selected is intended as a treatment or replication (i.e., the three tanks cited in line 286) within each treatment. Thanks.
In response: Thanks for your question. To avoid tank-derived deviations in the experiment, the sample for each biological replicate was taken from separate tank. Briefly, 90 domesticated turtles were randomly assigned to three groups (control, infection, purslane), each group includes 3 tanks, 10 turtles per tank. During sampling, 4 turtles were randomly sampled from 3 tanks in each group, each turtle is 1 biological replicate, so 4 turtles sampled from each group were set as 4 biological replicates.
